# Pentostatin Biosynthesis Pathway Elucidation and Its Application

Hongyu Zhang [1], Ran Liu [1], Tingting Lou [2], Pei Zhao [1] and Suying Wang [1,*]

[1] Tianjin Key Laboratory of Food Biotechnology, College of Biotechnology and Food Science, Tianjin University of Commerce, Tianjin 300134, China

[2] Animal, Plant and Foodstuffs Inspection Center of Tianjin Customs, Tianjin 300461, China

* Correspondence: wsying@tjcu.edu.cn

**Abstract:** Pentostatin (PNT), a nucleoside antibiotic with a 1,3-diazo ring structure, is distributed in several actinomycetes and fungi species. Its special structure makes PNT possess a wide spectrum of biological and pharmacological properties, such as antibacterial, antitrypanosomal, anticancer, antiviral, herbicidal, insecticidal, and immunomodulatory effects. Because of the promising adenosine deaminase inhibitory activity of PNT, its extensive application in the clinical treatment of malignant tumors has been extensively studied. However, the fermentation level of microbial-derived PNT is low and cannot meet medical needs. Because the biosynthesis pathway of PNT is obscure, only high-yield mutant screening and optimization of medium components and fermentation processes have been conducted for enhancing its production. Recently, the biosynthesis pathways of PNT in actinomycetes and fungi hosts have been revealed successively, and the large-scale production of PNT by systematic metabolic engineering will become an inevitable trend. Therefore, this review covers all aspects of PNT research, in which major advances in understanding the resource microorganisms, mechanism of action, and biosynthesis pathway of PNT were achieved and diverse clinical applications of PNT were emphasized, and it will lay the foundation for commercial transformation and industrial technology of PNT based on systematic metabolic engineering.

**Keywords:** pentostatin; nucleoside antibiotic; biosynthesis pathway; application; biological and pharmacological property



## 1. Introduction

There are numerous types of antibiotics, and most of the antibiotics currently used are mainly derived from microorganisms in nature [1–3]. Based on the different chemical structures, antibiotics are mainly divided into several groups, including polypeptides, aminoglycosides, tetracyclines, polyenes, nucleosides, polyethers, macrolides, and β-lactams [4]. Among them, nucleoside antibiotics are biologically active microbial secondary metabolites that are formed by a series of post-structural modifications of nucleosides or nucleotides, and they have shown remarkably broad biological activities, mainly including antibacterial, antitrypanosomal, anticancer, antiviral, herbicidal, insecticidal, and immunomodulatory effects [5–7].

PNT, as a member of nucleoside antibiotics, has attracted widespread interest due to its specific biological activity. PNT was first isolated from *Streptomyces antibioticus* in 1974, its specific structure also was analyzed, and PNT especially was first reported as a potent adenosine deaminase inhibitor [8]. Subsequently, more and more researchers became interested in PNT and its application. In 1975, Warner-Lambert applied for a patent, US 3923785, which provided a new fermentation process of PNT from *Streptomyces antibioticus*, but the yield was still very low [9]. In order to increase the yield, in 1982 the chemical synthesis of PNT was first achieved by Warner-Lambert [10]. Afterward, the chemical synthesis of PNT was studied in depth over the years, and significant results in terms of reaction

routes and production feasibility were achieved, but the chemical synthesis of PNT showed serious disadvantages, such as long synthetic routes, harsh reaction conditions, very low production efficiency, high cost of consumption, and difficulty in industrial production compared to microbial fermentation. With the rapid development of chromatography and structural analysis methods, PNT in *Streptomyces antibioticus* NRRL 3238, *Actinomadura* sp. ATCC 39365, *Aspergillus nidulans*, *Cordyceps militaris*, and *Cordyceps kyushuensis Kobayasi* was discovered successively [11–15]. Nowadays, the industrial production of PNT is mainly obtained from the fermentation broth of high-yield mutants. However, it is inefficient to obtain high-yield mutant strains by traditional breeding methods. Though the pharmaceutical and physiological importance of PNT is known, understanding of its biosynthesis pathways remains limited [16]. Therefore, it is necessary to study the biosynthesis mechanism of PNT for improving the yield of PNT through a further metabolic engineering strategy [17]. With the rapid development of synthetic biology and genome sequencing technology, the mechanism of PNT biosynthesis has been preliminarily elucidated in the *Streptomyces antibioticus* NRRL 3238, *Actinomadura* sp. ATCC 39365, and *Cordyceps militaris*, but the functions of some key genes in these gene clusters have been uncovered. Since PNT has been approved by the US Food and Drug Administration (FDA) as a commercial drug against hairy cell leukemia, and as attention increases, its extensive application in the clinical treatment of malignant tumors has been extensively studied [18,19]. Moreover, the therapeutic field of PNT has been expanding in recent years, including Waldenstrom's macroglobulinemia, Trypanosoma, and so on [20,21].

As a consequence, this review aims to cover all the key research areas of PNT in recent years. It will place particular emphasis on the resource microorganisms, mechanism of action, biosynthesis pathways, and diverse clinical applications of PNT. Furthermore, this review also has included a list of key functional genes of PNT production in the biosynthesis gene cluster, which will present the foundation and insights for the sustainable metabolic engineering of PNT production in multifarious organisms.

## 2. Resource Microorganisms of PNT

Actinomycetes and fungi are the main resource microorganisms of PNT. PNT was first isolated from the fermentation broth of *Streptomyces antibioticus* [8]. Purine nucleoside antibiotic 2′-amino-2′-deoxyadenosine was first isolated from the fermentation broth of *Actinomadura* sp. ATCC 39365 in 1979, and then PNT and 2′-chloropentostatin (2′-ClPNT) also were obtained subsequently [13,21,22]. The biosynthesis gene cluster of PNT and its associated product arabinofuranosyladenine (Ara-A) in *Streptomyces antibioticus* NRRL3238 was discovered in 2017 [13]. It was firstly reported in 1979 that *Aspergillus nidulans* could produce cordycepin (COR) and PNT in the meanwhile [15]. Subsequently, it was reported that the *Emericella nidulans* (the asexual form of *Aspergillus nidulans*) also could produce PNT [22]. With the development of genome sequencing technologies, some key functional genes in the biosynthesis gene cluster of PNT were discovered based on the genome sequence of the fungi *Cordyceps militaris* in 2017, and PNT and COR were also obtained in its fermentation broth [12]. By comparing with the PNT biosynthesis gene cluster of *Cordyceps militaris*, later studies demonstrated that four key genes involved in the biosynthesis of COR and PNT were identified in *Cordyceps kyushuensis Kobayasi* in 2019 [11]. In summary, the resource microorganisms of PNT and its associated products are shown in Table 1 and Figure 1.

**Table 1.** The resource microorganisms of PNT and its associated products.

| Species | Resource Microorganisms | Associated Products | References |
|---|---|---|---|
| Actinomycetes | *Streptomyces antibioticus* NRRL3238 | Ara-A | [13,21,22] |
| | *Actinomadura* sp. ATCC 39365 | 2′-ClPNT, 2′-amino-2′-deoxyadenosine | [8,13] |
| Fungi | *Aspergillus nidulans* | COR | [15] |
| | *Cordyceps militaris* | COR | [12] |
| | *Cordyceps kyushuensis Kobayasi* | COR | [11] |

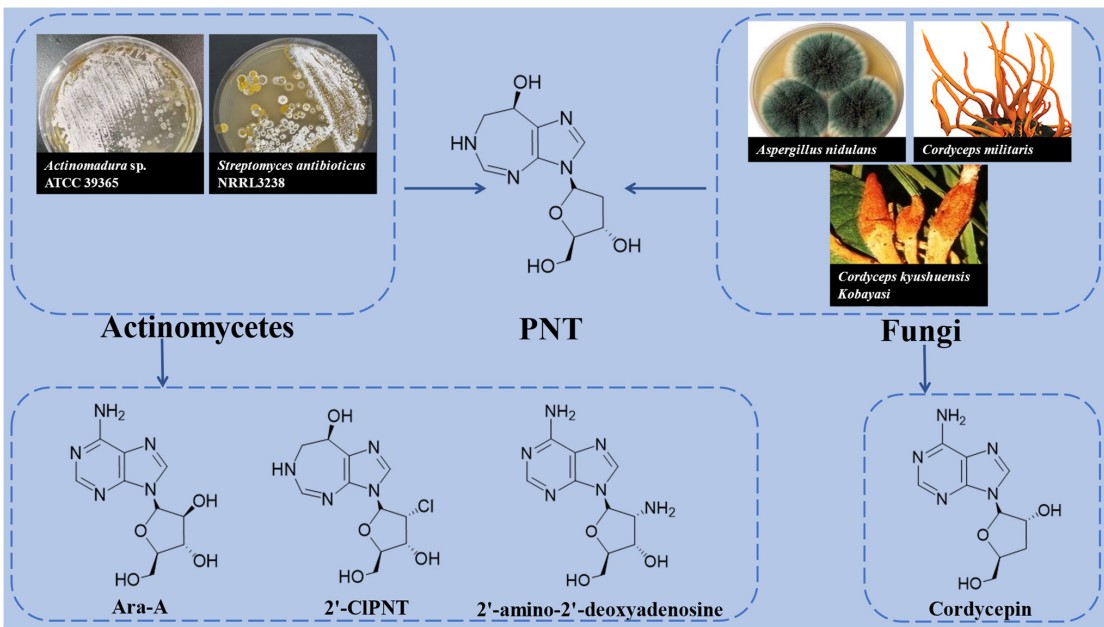

**Figure 1.** The structure and resource microorganisms of PNT and its associated products.

## 3. Action Mechanism of PNT

PNT is a transition of an intermediate in the adenosine deaminase reaction pathway [18]. Thus, PNT has the effect of inhibiting adenosine deaminase, which leads to the accumulation of intracellular adenosine or deoxyadenosine, resulting in blocked DNA synthesis in cells. Therefore, the FDA approved PNT as an injection, named Nipent, for the treatment of acute T-cell lymphoblastic leukemia, hair-cell leukemia, and chronic lymphoblastic leukemia in 1992 [23–25]. Its mechanism of action is as follows: adenosine deaminase (ADA), a key enzyme involved in the complementary pathway of purine metabolism, catalyzes the degradation of adenosine and deoxyadenosine to inosine and deoxyinosine in cells. ADA activity is abnormally elevated in the lymphocytes of patients with chronic lymphocytic leukemia, and PNT binds closely to ADA and inhibits its activity [26]. When ADA activity is inhibited, adenosine accumulates in large amounts, which is catalyzed by adenosine kinase and ribonucleotide reductase in lymphocytes to produce deoxyadenosine triphosphate (dATP). dATP can inhibit the activity of nucleotide reductase and produce feedback inhibition on lymphocyte proliferation and differentiation [27,28]. The action mechanism of PNT is shown in Figure 2.

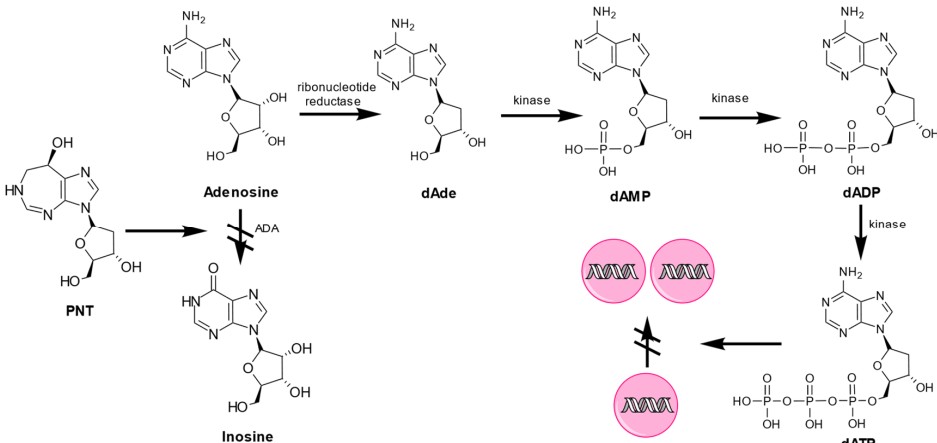

**Figure 2.** The action mechanism of PNT.

## 4. Biosynthesis Pathways of PNT

### 4.1. The PNT Biosynthesis Pathway in Streptomyces antibioticus NRRL 3238

In 1984, adenosine was shown to be a precursor of PNT in the *Streptomyces antibioticus* NRRL 3238 by an isotope feeding experiment, where glycine, adenine, and adenosine were chosen as feeding ingredients [29]. The one-carbon unit between purine ring C-6 and N-1 is derived from ribose C-1. It is speculated that the PNT biosynthesis pathway is closely related to histidine synthesis in primary metabolism [30]. Later, cell-free enzymatic reaction confirmed that 8′-keto-PNT was converted to PNT by NADPH reductase. Thus, 8′-keto-PNT is deduced to be an intermediate in the biosynthesis of PNT, which only preliminarily speculated the biosynthesis pathway of PNT, and the specific biosynthesis mechanism of PNT has not been elucidated [16].

With the rapid development of genome sequencing technology, it has played a very important role in promoting the discovery rate of novel biosynthesis gene clusters of natural products [31,32]. The whole genome sequencing of *Streptomyces antibioticus* NRRL3238 was completed in 2017 [13]. Given that the biosynthesis of PNT correlates with the initial steps of the L-histidine biosynthesis pathway, HisG from *Streptomyces coelicolor* A3(2), an enzyme required for the first step of the L-histidine pathway, was utilized as a probe to screen the *Streptomyces antibioticus* NRRL3238 genome data [16,33]. Therefore, *PenA*, which is a homologous gene of *HisG*, has been found. Based on this, the genomic library of *Streptomyces antibioticus* NRRL3238 was constructed, and the Cosmid12H4 containing the *PenA* gene was filtered out. The upstream and downstream sequence of the *PenA* gene in the Cosmid12H4 was analyzed, and the putative whole gene cluster was included in the Cosmid12H4. Then, Cosmid12H4 was introduced into *Streptomyces aureochromogenes* CXR14 for heterologous expression, which successfully produced PNT and its associated product Ara-A. Therefore, Cosmid12H4 contains the whole gene cluster for the biosynthesis of PNT and its associated products. Then, ten key functional genes in the biosynthesis gene cluster were knocked out one by one by PCR targeting, and heterologous expression of mutant gene cluster was performed in *Streptomyces aureochromogenes* CXR14, and the minimal boundary of the gene cluster was determined [34]. This gene cluster contains 10 genes from *PenA* to *PenJ*, with a total of 10.0 kb (Figure 3A and Table 2). The results confirmed that *PenA*, *PenB*, and *PenC* were related to the biosynthesis of PNT, and the genes of *PenD* to *PenJ* were related to the biosynthesis of Ara-A.

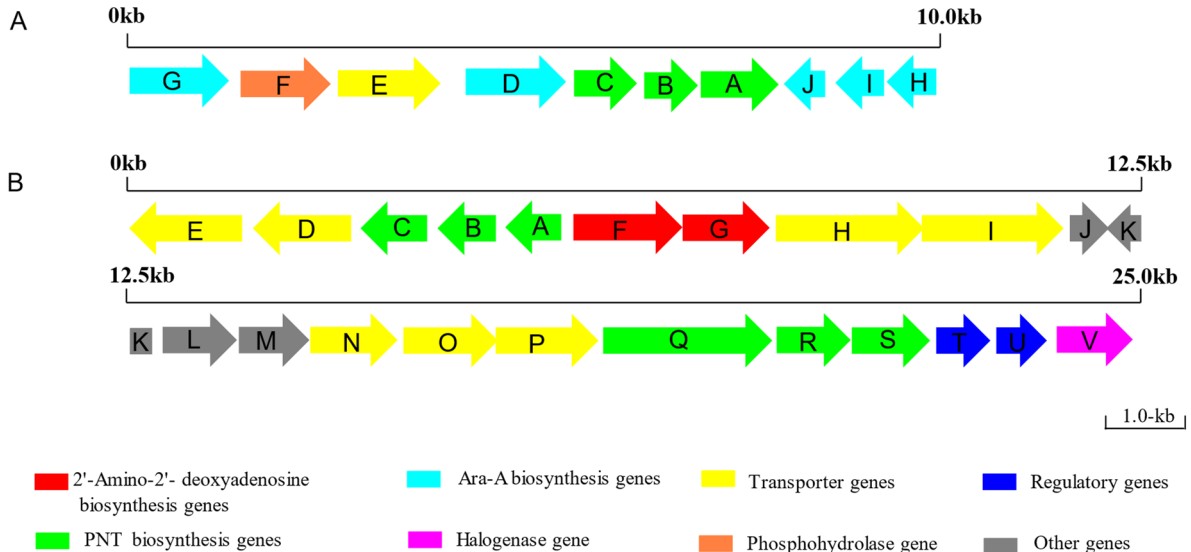

**Figure 3.** The biosynthesis gene cluster of PNT and associated products in the *Streptomyces antibioticus* NRRL 3238 and *Actinomadura* sp. ATCC39365. (**A**): The biosynthesis gene cluster in *Streptomyces antibioticus* NRRL 3238. (**B**): The biosynthesis gene cluster in *Actinomadura* sp. ATCC39365.

**Table 2.** The proposed functions of proteins in the biosynthesis gene cluster of PNT and Ara-A in the *Streptomyces antibioticus* NRRL 3238.

| Protein | Size (aa) | Proposed Function |
| --- | --- | --- |
| PenG | 387 | SAH hydrolase |
| PenF | 358 | Phosphohydrolase |
| PenE | 401 | membrane transport protein |
| PenD | 396 | SAH hydrolase |
| PenC | 247 | SAICAR synthetase |
| PenB | 234 | Short-chain dehydrogenase |
| PenA | 302 | ATP phosphoribosyl-transferase |
| PenJ | 161 | Dehydrogenase |
| PenI | 189 | Dehydrogenase |
| PenH | 213 | Dehydrogenase |

Previous metabolic labeling studies established that PNT biosynthesis is related to L-histidine biosynthesis [35]. About the histidine biosynthesis pathway, ATP/dATP and phosphoryl pyrophosphate (PRPP) were supposed to be starting compounds [16,29]. PenA (ATP phosphoribosyl-transferase) is proposed to regulate the initial step by the condensation of dATP/ATP and PRPP to produce compound 1. Subsequently, compound 1 is converted to compound 2 by the three enzymes HisI (phosphoribosyl-AMP cyclohydrolase), HisE (phosphoribosyl-ATP pyrophosphatase), and HisA (phosphoribosyl isomerase) from the histidine biosynthesis pathway. Compound 2 underwent a series of rearrangement and breakage reactions to generate the unstable intermediate compound 3 and compound 4. Compound 4 was catalyzed by the PenC (SAICAR synthetase) to generate compound 5 (5′-p -6′-keto PNT). This enzymatic reaction has never been previously reported. Then, compound 5 is dephosphorylated by a phosphatase to produce compound 6 (6′-keto-PNT). Finally, compound 6 is reduced by PenB (short-chain dehydrogenase) to accomplish compound 7 (PNT) biosynthesis. Previous studies have confirmed that SAH hydrolase catalyzes the reversible hydrolysis of SAH to form adenosine, and the precise reaction mechanism has been well established [36]. Studies of the pen gene cluster identified two SAH hydrolases (PenD and PenG) and three dehydrogenases (PenH, PenI, and PenJ) as suitable candidates for Ara-A biosynthesis. Subsequent results confirm that the biosynthesis of Ara-A is initiated by PenG (SAH hydrolase), which hydrolyzes SAH to adenosine, and PenD (SAH hydrolase) controls the reverse reaction [36]. Then, the adenosine is dehydrogenated at C′-2 to the intermediate 2′-keto adenosine by PenH (heteromeric dehydrogenase), PenI (heteromeric dehydrogenase), and PenJ (heteromeric dehydrogenase), followed by reduction to form the end product Ara-A (Table 2). Therefore, a biosynthesis pathway of PNT and its associated product Ara-A in the *Streptomyces antibiotic* NRRL 3238 was proposed (Figure 4A,B). Meanwhile, it also revealed that this gene cluster had an unusual protector–protégé strategy. PNT, as an ADA inhibitor, could protect its associated product Ara-A from deamination by ADA [13].

**Figure 4.** Biosynthesis pathway of PNT and associated products in the *Streptomyces antibiotic* NRRL 3238 and *Actinomadura* sp. ATCC39365 (modified from reference [13,14,37]. (**A**): biosynthesis pathway of PNT in the *Streptomyces antibiotic* NRRL 3238 and *Actinomadura* sp. ATCC39365. (**B**): biosynthesis pathway of Ara-A in the *Streptomyces antibiotic* NRRL 3238. (**C**): biosynthesis pathway of 2′-CLPNT in the *Actinomadura* sp. ATCC39365. (**D**): biosynthesis pathway of 2′- amino -2′-deoxyadenosine in the *Actinomadura* sp. ATCC39365. The red fonts indicate the biosynthesis pathway in *Streptomyces antibiotic* NRRL 3238. The blue fonts indicate the biosynthesis pathway in *Actinomadura* sp. ATCC39365. The dashed arrows indicate that this step is a speculative process. The solid arrows indicate that this step is a determination process.

### 4.2. The PNT Biosynthesis Pathway in Actinomadura sp. ATCC39365

*Actinomadura* sp. ATCC39365 has been previously characterized as a 2′-Cl PNT and 2′- amino-2′-deoxyadenosine producer, but this strain has never been reported to produce PNT [38,39]. In 2017, to address this question, Gao et al. investigated the metabolite profiles of *Actinomadura* sp. ATCC 39365 by liquid chromatography–mass spectrometry (LC-MS), and the results unquestionably demonstrated that *Actinomadura* sp. ATCC39365 is also a PNT producer [14]. Subsequently, three key genes (*PenA, PenB,* and *PenC*) of the PNT

biosynthesis pathway in *Streptomyces antibioticus* NRRL 3238 were employed as probes, and the genome sequence of *Actinomadura* sp. ATCC 39365 was screened, which revealed that the protein sequences of AdaA (ATP phosphoribosyltransferase), AdaB (short-chain dehydrogenase), and AdaC (SAICAR synthetase) from *Actinomadura* sp. ATCC39365 had high identities to PenC, PenB, and PenA from *Streptomyces antibioticus* NRRL 3238 with 59%, 71%, and 73%, respectively. Therefore, this target gene cluster including the gene of *AdaA*, *AdaB*, and *AdaC* is to be involved in the biosynthesis of PNT in *Actinomadura* sp. ATCC 39365. Based on this, the genome library of Cosmid3G12 containing the gene of *AdaA*, *AdaB*, and *AdaC* was obtained and introduced into *Streptomyces aureochromogenes* CXR14 for heterologous expression, which successfully produced PNT and associated products. Then, based on the Cosmid3G12 terminal sequence, by using bioinformatics analysis and gene knockout strategy, the minimal boundary of the Ada gene cluster was determined. Subsequently, the function of each gene in the Ada gene cluster was verified by using PCR targeting, and heterologous expression of the whole gene cluster was performed in *Streptomyces aureochromogenes* CXR14 [34]. This result confirmed that there was one gene cluster for the biosynthesis of PNT and its associated products; however, their biosynthesis pathways are independent. This gene cluster contains 13 genes from *AdaA* to *AdaM*, with a total of 14.4 kb. It is confirmed that *AdaA*, *B*, *C*, *E*, *K*, and *L* were related to the biosynthesis of PNT, and *AdaF*, *G*, *J*, and *M* were related to the biosynthesis of 2′-amino-2′-deoxyadenosine. However, the gene for the biosynthesis of 2′-ClPNT has not been discovered among 13 genes, it is necessary to further clarify how the chlorine is bound to the precursor to synthesize 2′-ClPNT. Subsequently, the biosynthesis gene cluster of PNT and associated products of *Actinomadura* sp. ATCC39365 was updated in 2019 [37]. The updated gene cluster was named Ade, from gene *AdeA* to *AdeV* with a total of 25.0 kb (Figure 3B). In addition, this study deduced that *AdeA*, *B*, *C*, *Q*, *R*, *S*, and *V* were related to the biosynthesis of 2′-ClPNT (Figure 4C).

Based on previous studies, the 2′-ClPNT/PNT biosynthesis pathway was deduced to be closely related to the pathway of primary L-histidine [13]. The results of this study are consistent with previous work on the PNT biosynthesis pathway in the *Streptomyces antibioticus* NRRL 3238. In *Actinomadura* sp. ATCC39365, the PNT biosynthesis pathway would begin with dATP/ATP and PRPP to form compound 1 by AdeC (ATP phosphoribosyltransferase) and AdeL (ATP phosphoribosyltransferase). Subsequently, compound 1 would be converted to compound 2 through the histidine pathway by three enzymes, HisI (phosphoribosyl-AMP cyclohydrolase), HisE (phosphoribosyl-ATP pyrophosphatase), and HisA (phosphoribosyl isomerase). However, it had been found that the PNT biosynthesis pathway contains a HisA homolog-AdaK (phosphoribosyl isomerase), which performed the same function as HisA. Under the action of AdeA (SAICAR synthetase), compound 2 was catalyzed to form compound 5 (5′-p -6′-keto PNT) via the intermediate compound 3 and compound 4 by a series of complex reactions. Compound 5 (5′-p -6′-keto PNT) was dephosphorylated by AdeM (hydrolase) to generate compound 6 (6′-keto-PNT). Finally, compound 6 (6′-keto-PNT) is dehydrogenated by AdeB (short-chain dehydrogenase) to generate compound 7 (PNT). Previous studies have demonstrated that adenosine is the direct precursor for the biosynthesis of 2′- amino -2′- deoxyadenosine [29], and adenosine is dehydrogenated to form 2′-keto adenosine, which is subsequently transaminated to produce the final product 2′-amino-2′-deoxyadenosine. In recent studies, the biosynthesis of 2′-amino -2′-deoxyadenosine is initiated by AdeJ (Nudix hydrolase), which catalyzes the hydrolysis of ATP to AMP. Then, the AdeM catalyzes intermediate AMP to form adenosine by dephosphorylating. The adenosine undergoes dehydrogenation to form the 2′-keto adenosine by AdeG (dehydrogenase). Finally, 2′-keto adenosine is catalyzed by AdeF (aminotransferase) to accomplish the biosynthesis of the 2′-amino-2′-deoxyadenosine. Another associated product, 2′-ClPNT, is a natural nucleoside analog containing chlorine in the *Actinomadura* sp. ATCC39365. However, its chlorination origin and the molecular mechanism of chlorination have been unclear for decades. Recently, it has been deduced that the initial substrate dATP undergoes dephosphorylation to generate 2′-dAMP. Sub-

sequently AdeV (halogenase) catalyzes the conversion from 2′-dAMP to 2′-Cl-2′-dAMP with $Fe^{2+}$ and $\alpha$-KG. Then, 2′-Cl-2′-dAMP and PRPP are catalyzed by AdeA, AdeB, AdeC, AdeQ (N,N-dimethylformamidase), AdeR (glucosamine-6-phosphate deaminase), and AdeS (ribokinase) to form 2′-Cl-2′-deoxyadenosine (Table 3) [37]. Finally, it undergoes dephosphorylation to form 2′-ClPNT. The proposed biosynthesis pathways of PNT and its associated products 2′-ClPNT and 2′-amino-2′-deoxyadenosine in the *Actinomadura* sp. ATCC39365 are shown in Figure 4A, C, and D. Similar to *Streptomyces antibiotic* NRRL 3238, a gene cluster exists in *Actinomadura* sp. ATCC39365 that can synthesize PNT, 2′-ClPNT, and 2′-amino-2′-deoxyadenosine separately. Meanwhile, a similar protector–protégé strategy to *Streptomyces antibiotic* NRRL 3238 exists in *Actinomadura* sp. ATCC39365, where 2′-ClPNT could inhibit ADA and prevent 2′-amino-2′-deoxyadenosine from being deaminated [14].

**Table 3.** The proposed functions of protein in the biosynthesis gene cluster of PNT, 2′-CLPNT, and 2′-amino-2′-deoxyadenosine in the *Actinomadura* sp. ATCC39365.

| Protein | Size (aa) | Proposed Function |
|---|---|---|
| AdeE | 479 | Cation/$H^+$ antiporter |
| AdeD | 402 | MFS transporter |
| AdeC | 295 | ATP phosphoribosyl-transferase |
| AdeB | 234 | Short-chain dehydrogenase |
| AdeA | 239 | SAICAR synthetase |
| AdeF | 425 | Aminotransferase |
| AdeG | 351 | Dehydrogenase |
| AdeH | 595 | ABC transporter, partial |
| AdeI | 592 | ABC transporter |
| AdeJ | 161 | Nudix hydrolase |
| AdeK | 257 | Phosphoribosyl isomerase A |
| AdeL | 288 | ATP phosphoribosyl-transferase |
| AdeM | 264 | Hydrolase |
| AdeN | 347 | ABC transporter substrate-binding protein |
| AdeO | 363 | Sugar ABC transporter permease |
| AdeP | 406 | Nucleoside ABC transporter |
| AdeQ | 700 | N, N-dimethylformamidase |
| AdeR | 303 | Glucosamine-6-phosphate deaminase |
| AdeS | 309 | Ribokinase |
| AdeT | 241 | Bacterial regulatory protein, gntR family |
| AdeU | 231 | Bacterial regulatory protein, luxR family |
| AdeV | 310 | 2OG-Fe (II) oxygenase |

*4.3. The PNT Biosynthesis Pathway in Cordyceps Militaris*

The biosynthesis pathway of PNT has been studied not only in Actinomycetes but also in fungi. It was previously reported that cordycepin and PNT were produced by *Aspergillus nidulans*, *Cordyceps militaris*, and *Cordyceps kyushuensis Kobayasi* [15,22]. To predict the gene cluster of PNT and the associated product COR in *Cordyceps militaris*, it is assumed that the biosynthesis genes may be conserved between the genomes of *Cordyceps militaris* and *Aspergillus nidulans* (Figure 5B), and then four linked genes were identified by genome-wide reciprocity analysis. Then, the four genes in *Cordyceps militaris* were designated as *Cns1*, *Cns2*, *Cns3*, and *Cns4*. BLAST analysis indicated that the proteins encoded by these four genes contain different conserved structures, such as the oxidoreductase/dehydrogenase domain in Cns1, the HDc family of metal-dependent phosphohydrolase domain in Cns2, and the N-terminal nucleoside/nucleotide kinase (NK) and C-terminal HisG domains in Cns3. Cns4 is a putative ATP-binding cassette type of transporter (Table 4). This gene cluster contains four genes from *Cns1* to *Cns4*, with a total of 10.3kb (Figure 5A). To determine whether this gene cluster is responsible for cordycepin and PNT biosynthesis, the serial gene deletions and complementation of *Cns1–Cns4* in *Cordyceps militaris* were carried out. To further validate the function of the genes, the heterologous gene expression in *Saccharomyces cerevisiae* and *Metarhizium robertsii* were

performed. First, the cDNA sequences of *Cns1–Cns3* and *Cns1–Cns2* were cloned, and then the two gene cluster fragments were transferred into *Saccharomyces cerevisiae*. The *Cns1–Cns3* gene cluster fragment was transferred into *Metarhizium robertsii*. These results confirmed that *Cns1–Cns3* are responsible for COR biosynthesis and the essential roles of *Cns1* and *Cns2* genes in COR production. In addition, to verify the function of *Cns3*, the heterologous expression of *Cns3* in *Metarhizium robertsii* and *Cordyceps bassiana* was employed. It is confirmed that the *Cns3* knockout mutant strain could not produce PNT. In addition, they further attempted a partial cDNA sequence, encoding either the NK or the HisG domain of Cns3 to complement the knockout mutant *Cordyceps militaris*. The results revealed that the complementation with the HisG sequence was able to produce PNT, but the NK domain did not.

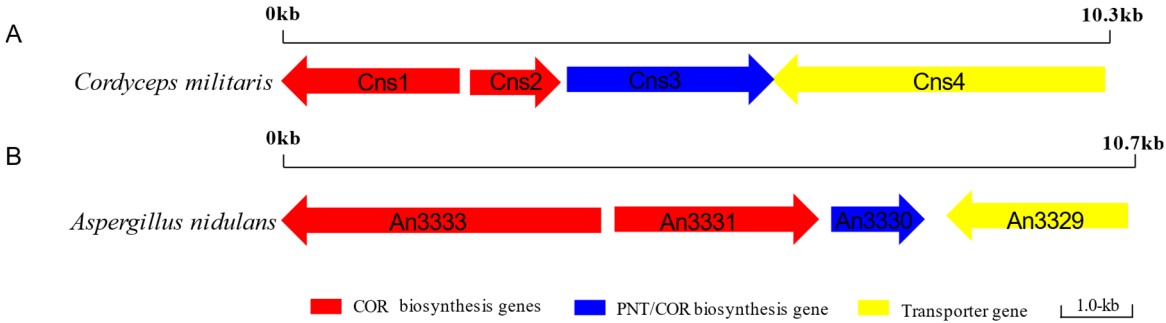

**Figure 5.** (**A**): the biosynthesis gene cluster of PNT and COR in the *Cordyceps militaris*. (**B**): the biosynthesis gene cluster of PNT and COR in the *Aspergillus nidulans*.

**Table 4.** The proposed functions of proteins in the biosynthesis gene cluster of PNT and COR in the *Cordyceps militaris*.

| Protein | Size (aa) | Proposed Function |
|---------|-----------|-------------------|
| Cns1 | 792 | Oxidoreductase/dehydrogenase |
| Cns2 | 345 | Metal-dependent phosphohydrolase |
| Cns3 | 871 | Phosphoribosyltransferase |
| Cns4 | 1364 | ATP-binding cassette (ABC) transporter |

In summary, the biosynthesis of PNT is initiated by the HisG domain of Cns3 from adenosine and PRPP. However, the NK domain of Cns3 converts precursor adenosine to 3′-AMP [40,41]. Subsequently, the conversion of 3′-AMP to 2′-C-3′-dA is catalyzed by Cns2, and 2′-C-3′-dA will be further converted to COR by Cns1 [42]. Cns4 (ATP-binding cassette transporter) is a transport protein, which mainly transports PNT out of the cell. Finally, the biosynthesis pathways of PNT and its associated product in *Cordyceps militaris* are shown in Figure 6. Although PNT could be produced in eukaryotes in previous studies, the PNT-dependent protector–protégé strategy had not been reported in eukaryotes. However, the subsequent result confirmed that PNT protected COR from deamination in the *Cordyceps militaris* similar to *Streptomyces antibiotic* NRRL 3238. It was the first time the protector–protégé strategy in eukaryotes was reported [12].

**Figure 6.** Biosynthesis pathway of PNT and COR in the *Cordyceps militaris* (modified from reference [12]). (**A**): biosynthesis pathway of PNT. (**B**): biosynthesis pathway of COR. The solid arrows indicate that this step is a determination process.

## 5. Applications of PNT

### 5.1. Treatment of the Hairy Cell Leukemia

Hairy cell leukemia (HCL) is a relatively unusual chronic B-cell lymphoproliferative disorder that clinically presents with bone marrow and the spleen being infiltrated [43,44]. Interferon-$\alpha$ (IFN-$\alpha$) was the first effective treatment for HCL. Based on its effect on HCL, IFN-$\alpha$ was granted by the FDA [45]. However, with the discovery of new natural drugs, PNT has attracted increasing attention for the treatment of HCL [46]. After comparing the treatment effects on HCL, PNT was significantly more effective than IFN-$\alpha$ [47,48]. For several years, subcutaneous IFN-$\alpha$ was the drug of choice for the treatment of HCL, but it was later replaced by PNT. PNT was the first purine analog to undergo extensive testing as an anticancer agent and the first medicine to receive FDA approval for a treatment indication [43]. As a first-line treatment for HCL, PNT therapy is very effective, and PNT has offered good long-term prospects for HCL patients in recent years [49,50]. Different researchers have simultaneously demonstrated that PNT leads to complete remission rates of over 75% and 10-year overall survival rates of over 80% in HCL patients [50,51]. During the treatment of HCL, PNT is generally well tolerated due to lymphocytopenia and inhibition of adenine deaminase [52].

### 5.2. Treatment of the Chronic Lymphoblastic Leukemia

Lymphocytic leukemia is a common tumor of the blood system, which can be divided into acute lymphocytic leukemia and chronic lymphocytic leukemia. Acute lymphocytic leukemia is common in children, and chronic lymphocytic leukemia is common in the elderly [53]. Chronic lymphocytic leukemia (CLL) is the most prevalent B-cell lymphoproliferative disorder. The B-cell of CLL is characterized phenotypically by co-expression of the B-cell proteins CD19, CD20, CD22, and CD5 [54].

The first evidence for the anti-CLL effects of PNT appeared in reports and trials [55]. Great progress has been made in the treatment of CLL, and the most widely used drug for CLL is Fludarabine. Subsequent studies demonstrated that PNT was also clinically active against CLL and less toxic than Fludarabine [49]. PNT in conjunction with cyclophosphamide and rituximab as first-line therapy for lymphocytic leukemia proved to be effective in inducing remission with acceptable toxicity in older and younger patients [56]. An experimental protocol was designed using PC chemotherapy in combination with the

fully human anti-CD20 monoclonal antibody, and the protocol specifically designed for untreated CLL patients aged over 65 years showed that the combination of PNT with cyclophosphamide was better tolerated and less myelosuppressive than fludarabine [53]. PNT was an effective and well-tolerated nucleoside therapy for CLL patients with a relapsed/refractory (R/R) condition after extensive pretreatment in 120 patients with a mean age of 64 years [25].

### 5.3. Treatment of Waldenstrom's Macroglobulinemia

Waldenstrom's macroglobulinemia (WM) is an indolent lymphoma, and commonly used therapeutic drugs mainly include alkylating agents, rituximab, and nucleoside. Nucleosides, such as fludarabine and cladribine, are considered to be the appropriate first-line drugs for WM [57,58]. However, PNT is effective in several lymphoid malignancies. Recent studies have demonstrated the efficacy and safety of PNT in combination with other drugs to treat WM better than a single drug. A phase II trial was initiated to determine the safety and efficacy of a regimen containing PNT, cyclophosphamide, and rituximab in patients with WM. Twenty-one patients received PER as first-line therapy. The results demonstrated that the combined use of PNT, cyclophosphamide, and rituximab was a safe and effective treatment for WM [20,59].

### 5.4. Inhibition of the Trypanosoma

*Trypanosoma evansi* and *Trypanosoma cruzi* pose a threat to human and animal life. Therefore, it is crucial to study drugs that kill or inhibit trypanosomes. Chagas disease was a neglected disease, which was caused by the protozoan parasite *Trypanosoma cruzi* [60]. *Trypanosoma evansi* is a flagellated protozoan that can infect a variety of hosts and cause devastating diseases in horses. It is the most popular pathogenic Trypanosoma in tropical and subtropical regions of the world [61]. Clinically, the infection manifests as rapid weight loss, varying degrees of anemia, intermittent fever, edema of the hind limbs, progressive weakness, and dyskinesia, which eventually leads to death [62].

In recent years, research on the inhibitory effect of PNT on trypanosomes was conducted on mice. It was found that 2 mg/kg of PNT was effective in suppressing *Trypanosomes evansi*, and the livers and kidneys of the mouse model were damaged. Subsequently, the results demonstrated the effectiveness and low toxicity of the treatment for the mouse model [63,64]. In addition, by studying the dosage of PNT alone or in combination with COR to inhibit or kill *Trypanosoma cruzi* in vivo and in vitro, it was found that the combined use of PNT and COR has an obvious effect on killing *Trypanosoma cruzi* in vitro, but it has no therapeutic effect in the in vivo experiment of the infected *Trypanosoma cruzi* mouse model [21]. At the same time, it was found that these treatments could regulate purinergic enzymes of the mouse model, which could help to reduce inflammatory damage to the heart [65].

### 6. Conclusions

The nucleoside antibiotic PNT has been very effective in the treatment of tumors, especially in the treatment of hairy cell leukemia. In recent studies, it has turned out that the application of PNT has become more widespread, and as a result the usage of PNT might be increased. According to the available studies, the synthesis of PNT includes chemical synthesis and biosynthesis. However, the chemical synthesis of PNT has major disadvantages compared to biosynthesis. Therefore, PNT is now produced in large quantities mainly through fermentation. Initially, the way to increase the yield of PNT was mainly by optimizing the composition of the culture medium and the fermentation parameters. With continuous advances in high-throughput sequencing and synthetic biology technology, the biosynthesis gene clusters of PNT and its associated products have been elucidated in recent years in three typical strains. The functions of most genes in the gene cluster have been verified. Thus, the mechanism of PNT biosynthesis has been uncovered, and the biosynthesis of PNT can be regulated in the hope of obtaining

mutants with high PNT yield. Although the functions of most genes mentioned in the PNT biosynthesis gene cluster have been verified, there are still a small number of genes with obscure functions, only predicted by using bioinformatics tools, so further investigation is still required to complete the verification of all gene functions.

Based on the discussion, we summarized the following prospects and future trends. First of all, to explore the regulators of PNT biosynthesis and analyze its regulatory mechanism, to lay the foundation for the construction of a high-yielding PNT mutant. A metabolic engineering approach can be employed to modify the strains and construct high-yield PNT mutants with stable genetic traits based on the clear biosynthesis and regulation mechanism. For example, the functions of regulatory genes of *AdeT* and *AdeU* in the biosynthesis gene cluster of PNT in the *Actinomadura* sp. ATCC39365 remain unclear, both of which may regulate PNT synthesis. Therefore, after function identification of *AdeT* and *AdeU*, by knocking out the negative regulatory genes or overexpressing positive regulatory genes, the high-yielding mutant strains of PNT can be obtained. The global regulatory genes also can be predicted by whole-genome sequencing, and multigene knockout mutants can be constructed. In addition, microbial cell factories can be constructed in mature chassis cells based on a heterologous expression strategy for directed biosynthesis of PNT, for example, *Streptomyces lividans*, *Streptomyces coelicolor*, and *Streptomyces aureochromogenes* CXR14 can be employed as the chassis cells for the larger quantities' production of PNT. All in all, this review provides insight for researchers who will further investigate the biosynthesis of PNT for commercial transformation.

**Author Contributions:** H.Z., R.L. and T.L.: writing—original draft, conceptualization, and visualization; P.Z.: writing—review and editing; S.W.: supervision and funding acquisition. All authors have read and agreed to the published version of the manuscript.

**Funding:** This project is funded by the 2021 Tianjin Graduate Research and Innovation Project under grant number 2021YJSS296.

**Institutional Review Board Statement:** Not applicable.

**Informed Consent Statement:** Not applicable.

**Data Availability Statement:** Not applicable.

**Conflicts of Interest:** The authors declare no conflict of interest.

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
