# Peer review of "Pentostatin Biosynthesis Pathway Elucidation and Its Application"

_fermentation, doi:10.3390/fermentation8090459_

Round 1

Reviewer 1 Report

Dear Editor-in chief;

The article entitled (Pentostatin biosynthesis pathway elucidation and its applications) it is a good job, but the article requires minor corrections to be suitable for publishing. The following comments should be taken in consideration:

1. There are some typing mistakes, for example:-

There are many spelling errors such as the lack of a space between two words or more than a space in another place.

2. The purity of most figures is very poor.

3. When citing tables and figures within the text of the manuscript, it is preferable that the whole word be in the form of lowercase letters.

4. It is best to update the list of references and add some references in the current year 2022

Thank you very much.
Best wishes

Gaber O. Moustafa

Reviewer 2 Report

This is timely and detailed review of the current state of knowledge in the field of biosynthesis and biological activity of pentostatin, medically important nucleoside natural product. A few comments and suggestions on how to improve the presentation of results are given below.

Table 1, there is a typo - Actionmycetes

L. 96 - transition, not transit analog

Fig. 2 - if Ada converts adenosine into inosine, then the arrow on the scheme should go in opposite direction. Next, ribonucleotide reductase is not laleled on the scheme, while kinase is. Please check this figure for consistent labeling of  its elements.

Figures 4 and 6 largely overlap; I suggest authors to combine these figures into one; this will help to understand common and distinct steps of pentostatin production in Streptomyces and Actinomadura.

The same is true about figures 3 and 5 - by combining biosynthetic gene clusters in one figure will greatly aid the understanding of the material, without shuttling between different parts of the article.

Conclusions section should be re-written to keep the focus on the main topic of the review, which is an access to larger quantities of pentostation via metabolic engineering. I encourage authors to share their thought more concretely as to what genes can be used to improve pentostatin production.
